# Bullous Pemphigoid Develops Independently of DAP12

**DOI:** 10.3390/biom15111549

**Published:** 2025-11-05

**Authors:** Manuela Pigors, Sabrina Patzelt, Maëlys Brudey, Shirin Emtenani, Stanislav Khil’chenko, Mayumi Kamaguchi, Niklas Reichhelm, Melissa Parker, Katja Bieber, Ralf J. Ludwig, Enno Schmidt

**Affiliations:** 1Lübeck Institute of Experimental Dermatology, University of Lübeck, 23562 Lübeck, Germany; 2Incyte Research Institute, Wilmington, DE 19803, USA; 3Department of Dermatology, Allergology and Venerology, University of Lübeck, 23538 Lübeck, Germany

**Keywords:** autoimmune blistering disease, basement membrane zone, type XVII collagen, parsaclisib, PI3K, TREM1, TREM2, tyrobp

## Abstract

The adaptor molecule DNAX-activating protein of 12 kDa (DAP12) is broadly expressed in innate immune cells, but its role in autoimmunity remains unclear due to its dual regulatory functions. We investigated the contribution of the DAP12 pathway to bullous pemphigoid (BP), the most common autoimmune blistering disease, using a mouse model induced by transfer of anti-type XVII collagen (Col17) IgG. Repeated anti-Col17 IgG injections over 12 days produced comparable disease activity in DAP12-deficient and wildtype mice (*n* = 17/group), indicating that disease induction occurs independently of DAP12 signaling. Flow cytometry and immunofluorescence analysis of lesional skin further revealed a strong upregulation of the DAP12-associated triggering receptors expressed on myeloid cells (TREM) 1 in wildtype BP lesions, whereas TREM2^+^ cell frequencies in anti-Col17 IgG-treated wildtype and DAP12 knock-out animals were significantly lower than in healthy controls. Additional flow cytometry analysis demonstrated altered inflammatory infiltrates with notably reduced frequencies of Siglec-f^+^ eosinophils in DAP12-deficient vs. wildtype lesional skin. In addition, pharmacological inhibition of PI3Kδ, a downstream kinase of the DAP12/TREM pathway, did not affect disease progression in anti-Col17 IgG-induced BP. Collectively, these findings indicate that while DAP12 signaling modulates local immune cell composition, the DAP12/TREM1/2-axis does not influence overall disease activity in experimental BP.

## 1. Introduction

Pemphigoid diseases comprise a group of rare yet persistent autoimmune blistering disorders characterized by autoantibody-mediated damage of junctional adhesion molecules in the basement membrane zone, leading to subepithelial blister formation and inflammation affecting the skin and mucous membranes [1]. Among them bullous pemphigoid (BP) is the most frequent, with an estimated global annual incidence of 8.2 cases per million, and the incidence rates continue to rise [2,3]. In Central Europe and the United States, the reported rates have reached 20 cases per million [4]. Clinically, BP manifests with tense bullae on erythematous or normal skin accompanied by severe pruritus [1,3], while mucosal involvement is relatively rare [5]. Pathogenic autoantibodies can be directed against the intracellular protein BP230 in approximately half of patients, but the primary target antigen is type XVII collagen (Col17, also known as BP180), which is recognized in nearly all patients [3,6]. Additional immunopathological characteristics include the deposition of IgG and complement C3 at the cutaneous basement membrane zone, subepidermal cleavage, and inflammatory infiltrate in the upper dermis mainly composed of eosinophils and T cells, and, to lesser extent, neutrophils [7]. Despite systemic corticosteroids and immunosuppressants being the mainstay of treatment, as well as the availability of emerging biologics such as the anti-IL-4/IL-13 receptor blocker dupilumab, achieving durable disease control remains challenging, and long-term use of former agents is associated with substantial morbidity and mortality [3,8,9]. These limitations underscore the unmet need for deeper mechanistic insights to guide the development of safer, more targeted therapies.

One potential pathway of interest for pemphigoid diseases is mediated by the signaling adaptor molecule DNAX-activating protein of 12 kDa (DAP12; encoded by TYRO protein tyrosine kinase-binding protein [tyrobp]). DAP12 is a member of type I transmembrane adapter proteins, which is broadly expressed across innate immune cells, including neutrophils, macrophages, dendritic cells, basophils, eosinophils, and natural killer cells [10,11,12]. Moreover, some T and B cell subsets were reported to express DAP12 under inflammatory conditions [13]. The effector functions of DAP12 are mediated through its immunoreceptor tyrosine-based activation motif (ITAM) domain, which is critical for immunoreceptor signaling [10,13,14]. DAP12 associates with more than 20 immunoreceptors, as well as other receptor families (e.g., cytokine receptors, integrins, adhesion molecules), in both mice and humans, mediating either activating or inhibitory signals [10,14,15]. In particular, the triggering receptors expressed on myeloid cells (TREMs) are broadly on their surface and have been shown to associate with DAP12, whereby TREM1 was reported to potentiate inflammatory responses and TREM2 to drive anti-inflammatory effects [16]. Upon crosslinking of DAP12-associated receptors, such as TREMs, intracellular signaling is initiated through the recruitment and activation of spleen tyrosine kinase (SYK) [17]. This leads to subsequent activation of phosphatidylinositol 3-kinase (PI3K), which in turn triggers the serine/threonine kinase AKT and promotes signaling through the mechanistic target of rapamycin (mTOR) pathway. Both SYK and PI3K isoforms have already been implicated in pemphigoid disease and are under investigation as prospective therapeutic targets for autoimmune blistering diseases [18,19].

Given the central role of myeloid cells and complement-driven inflammation in BP, as well as the dual regulatory functions of DAP12 signaling, understanding its contribution to BP pathogenesis is of considerable interest. In this study, we investigated the role of the DAP12 axis in a murine model of BP induced by transfer of anti-Col17 IgG targeting the murine extracellular non-collagenous domains 14–1 (NC14–1), which, in addition to IgG targeting the NC15A domain, the murine homologue of the human immunodominant NC16A domain, are capable of triggering an inflammatory response and tissue injury in the skin [20]. By dissecting the functional impact of DAP12 deficiency on disease development and inflammatory infiltrates, we sought to clarify whether this pathway contributes to autoantibody-mediated tissue damage in BP and to evaluate its potential as a novel therapeutic target for treatment of the disease.

## 2. Materials and Methods

### 2.1. Ethics Approval

Animal experiments were approved by the Schleswig-Holstein Ministry of Energy Transition, Agriculture, Environment, Nature and Digitalization (protocol codes 35-3/19 (approved on: 4 June 2019) and 15-3/20 (approved on: 3 April 2020)).

### 2.2. Mice

Animals were housed under specific pathogen-free conditions with a 12 h light-dark cycle and fed acidified drinking water and standard chow ad libitum. DAP12 knock-out (DAP12^−/−^) mice (B6.129P2-Tyrob^ptm1lll^; Taconic Biosciences, Germantown, NY, USA) were backcrossed to C57BL/6J mice (Janvier, Le Genest-Saint-Isle, France). Genotyping was performed by TransnetYX (Cordova, TN, USA) using genomic DNA from ear biopsies and previously published primers [21]. Wildtype littermates served as controls. Sex-matched mice aged 9–37 weeks with comparable age distributions across groups were used. Anesthesia was induced by intraperitoneal (i.p.) administration of a mixture of ketamine (100 mg/kg; Sigma Aldrich, Steinheim, Germany) and medetomidine (1 mg/kg; Orion Pharma, Espoo, Finland). Sedation was reversed with atipamezole (i.p. 5 mg/kg; Orion Pharma). For pharmacological modulation with parsaclisib (Incyte, Wilmington, DE, USA), C57BL/6J mice aged 15–34 weeks, matched for age and sex, were obtained from in-house colonies held at the University of Lübeck animal facility. Anesthesia was induced with i.p. ketamine (100 mg/kg) and xylazine (15 mg/kg, Sigma Aldrich).

### 2.3. Antibody Transfer-Induced BP Mouse Model and Treatment

Mice were injected subcutaneously (s.c.) with 10 mg of total IgG targeting the NC14–1 domains of Col17 every other day from Day 0 to Day 10 to induce BP [20]. Control mice received s.c. 10 mg normal rabbit IgG. For pharmacological inhibition of PI3Kδ, mice were treated with parsaclisib (INCB050465, Incyte) [22] at 3 mg/kg twice daily by oral gavage from D 0 to Day 11. Parsaclisib was freshly prepared in vehicle consisting of 95% (*v*/*v*) methylcellulose solution (0.5% (*w*/*v*) and 5% (*v*/*v*) N,NDimethylacetamide (Sigma Aldrich). The percentage of affected body surface area was assessed under anesthesia in a blinded fashion every four days (Day 4, 8, 12). Disease scoring was performed as described [23]. At the endpoint, mice were sacrificed under anesthesia by cervical dislocation, and skin biopsies were collected for further analyses.

### 2.4. Direct and Indirect Immunofluorescence (IF) Staining

Direct IF staining was performed on 6-µm-thick cryosections from perilesional skin and site-matched controls to detect tissue-bound IgG and C3 deposits. FITC-labeled donkey anti-rabbit IgG (Jackson ImmunoResearch, West Grove, PA, USA) or goat anti-mouse C3 (MP Biomedicals, Santa Ana, CA, USA) were used as secondary antibodies. IgG and C3 signals were semi-quantified in a blinded fashion using a subjective scoring system from 0 (no signal) to 4 (highest signal intensity) as previously described [20]. Indirect IF staining on 6-µm-thick cryosections was performed following established protocols [24,25]. Primary antibodies included TREM1 (efluor660; ThermoFisher, Waltham, MA, USA) and Ly6G (Alexa 488; Biolegend, San Diego, CA, USA). Slides were mounted using DAPI Fluoromount-G^®^ (SouthernBiotech, Birmingham, AL, USA). Stained skin sections were visualized with a Keyence microscope (BZ-9000 or BZ-X810 series, Keyence GmbH, Neu-Isenburg, Germany).

### 2.5. Histological Analysis

Hematoxylin & eosin staining as well as histological analysis of lesional skin biopsies and site-matched controls, along with scoring, was performed in a blinded manner according to standard protocols as described [20].

### 2.6. Flow Cytometry

Sample preparation and flow cytometric analysis of lesional skin biopsies and site-matched controls were performed as described [20]. Single-cell suspensions were stained with fluorochrome-conjugated antibodies against the following surface markers: CD11b (PE-Cyanine 7), CD11c (Pacific Blue), CD3ε (PE-Dazzle-594), CD45 (Brilliant Violet 510), CD170/Siglec-F (Alexa Fluor 700), F4/80 (PerCP), Ly-6G (APC), NK1.1 (PE) (all Biolegend), TREM1 (efluor660, Thermo Fisher), and TREM2 (FITC, Biorad, Feldkirchen, Germany). Samples were acquired on a Cytek Aurora flow cytometer (Cytek Biosciences, Fremont, CA, USA) and analyzed using FlowJo software (version 10, Tree Star, Ashland, OR, USA). Gating was carried out as described [26]. Dendritic cells were defined as CD45^+^CD11c^+^, eosinophils as CD45^+^ Siglec-f^+^, macrophages (within the CD45^+^ population) as CD11b^+^F4/80^+^, natural killer cells as CD45^+^ NK1.1^+^, neutrophils (within the CD45^+^ population) as CD11b^+^Ly6G^+^, and T cells as CD45^+^CD3^+^. TREM1^+^ and TREM2^+^ subpopulations were analyzed within the CD45^+^ gate.

### 2.7. Statistical Analyses

Statistical analyses were performed using GraphPad Prism (Version 10, GraphPad Software, San Diego, CA, USA). Data are presented as mean ± standard deviation, and *p* < 0.05 was considered statistically significant.

## 3. Results

### 3.1. Loss of DAP12 Does Not Impact Disease Activity in Anti-Col17 IgG-Induced BP

To investigate the role of DAP12 in BP, we utilized a recently established mouse model induced by transfer of IgG targeting the murine Col17 extracellular domains NC14–1 [20]. Throughout the study period, disease activity, as assessed by body surface area involvement, was comparable between the DAP12^−/−^ and wildtype mice (*n* = 17/group) (Figure 1A,B). Similarly, histopathological features of lesional skin biopsies, including subepidermal blister formation and inflammatory cell infiltration, were comparable between the two groups (*n* = 16/group) (Figure 1C,D). Deposition of IgG and (Figure 1E,F) complement C3 (Figure 1G,H) along the dermal-epidermal junction was also equivalent in both groups (*n* = 16/group). No pathological changes were observed in DAP12^−/−^ mice treated with normal rabbit IgG.

### 3.2. Dysregulation of DAP12-Associating Receptors TREM1 and TREM2 Has No Impact on BP Pathogenesis

Although disease activity did not differ between DAP12^−/−^ and wildtype mice, we further examined lesional skin biopsies by flow cytometry to characterize potential molecular alterations underlying disease pathogenesis. Anti-Col17 IgG treatment induced marked upregulation of the DAP12-associated receptor TREM1 in wildtype mice, whereas expression remained low in DAP12^−/−^ mice and untreated wildtype controls (*p* < 0.001; *n* ≤ 14/group) (Figure 2A). Indirect IF analysis confirmed the pronounced reduction in TREM1 expression in anti-Col17 IgG-treated DAP12^−/−^ mice compared with wildtype controls, particularly in Ly6G+ neutrophil-rich regions (Figure 2B). In contrast, TREM2^+^ cells were significantly decreased in anti-Col17 IgG-treated wildtype mice as well as in both DAP12^−/−^ groups, compared with healthy skin from untreated mice, in which moderate TREM2 expression was detectable (*p* < 0.001; *n* ≤ 14/group) (Figure 2A).

### 3.3. Changes in the Inflammatory Infiltrate Composition in DAP12-Deficient Skin Do Not Affect BP

Since DAP12 and its associated TREM1 and TREM2 receptors are expressed on multiple immune cell populations, we next examined the composition of inflammatory infiltrates in lesional and site-matched murine control skin (*n* ≤ 14/group). CD11b^+^Ly6^+^ neutrophil frequencies were comparable in anti-Col17 IgG-treated wildtype and DAP12^−/−^ skin, consistent with unchanged disease severity (Figure 3A). In contrast, Siglec-f^+^ eosinophils were significantly reduced in anti-Col17 IgG-treated DAP12^−/−^ skin compared with BP skin in wildtype mice (*p* < 0.01) (Figure 3B). We also assessed skin-resident immune cells that are typically present under physiological conditions [27,28]. No DAP12-dependent differences were observed; however, both genotypes exhibited decreased CD3^+^ T cell frequencies following anti-Col17 IgG-treatment (*p* < 0.01) (Figure 3C). Furthermore, CD11c^+^ dendritic cells (*p* < 0.0001) and NK1.1^+^ natural killer cells (*p* < 0.01) were markedly increased in anti-Col17 IgG and normal rabbit IgG-treated groups compared with untreated controls (Figure 3D,E). By contrast, the proportions of CD11b^+^F4/80^+^ macrophages remained unchanged across all groups (Figure 3F).

### 3.4. Pharmacological Modulation of PI3Kδ Does Not Affect Disease Activity in Anti-Col17 IgG-Induced BP

After analyzing the role of DAP12 and TREM receptors in experimental BP, we next tested the effect of pharmacological inhibition of PI3Kδ, a downstream signaling mediator of DAP12, using parsaclisib, a well-characterized inhibitor demonstrating high potency and selectivity of the δ isoform of PI3K. Mice received prophylactic treatment with the PI3Kδ inhibitor in parallel with BP induction by anti-Col17 IgG (Figure 4A). Consistent with the DAP12-independent disease development observed in the DAP12^−/−^ mice, parsaclisib treatment did not alter the extent of the affected body surface area compared with vehicle-treated controls (*n* = 16/group) (Figure 4B). Likewise, histopathological assessment of skin lesions revealed no significant differences between the two groups (*n* = 8/group) (Figure 4C,D).

## 4. Discussion

To date, the role of DAP12 in autoimmunity remains controversial. While DAP12 has been shown to protect against autoimmunity in DAP12^−/−^ mice [21], and inhibition of the DAP12 signaling pathway reduces neutrophil activation in rheumatoid arthritis [29], paradoxically, other studies, have reported increased susceptibility to experimentally induced diabetes and autoimmune encephalomyelitis in DAP12^−/−^ mice [30,31]. In line with the mild osteopetrosis and thalamic hypomyelinosis observed in the DAP12^−/−^ mice [32], loss-of-function mutations in *TYROBP* (encoding DAP12) are associated with Nasu-Hakola disease, which is characterized by presenile dementia and bone cysts [33]. Collectively these findings suggest that DAP12 exerts diverse functions in the central nervous system, bone remodeling, and immune regulation. Although its role in autoimmunity is debated, the pleiotropic functions of DAP12 in positive and negative regulation of leukocyte responses suggest that its targeted modulation could represent a therapeutic strategy for autoimmune conditions, such as pemphigoid disease. To specifically address this, we evaluated the role of DAP12 in experimental BP using a published DAP12^−/−^ mouse model [21]. Following antibody transfer of anti-Col17 IgG, no phenotypic or immunopathological differences in disease activity were observed between DAP12^−/−^ and their wildtype littermates. No additional pathology was noted in DAP12^−/−^ mice, although a systematic evaluation of other organs was not performed.

Given the absence of overt effects of DAP12 deficiency in our BP model, we next asked whether associated signaling partners might be affected or compensate for its absence. We have previously demonstrated that anti-Col17 IgG-mediated BP is partially driven by complement C5a/C5aR1 and Fc gamma receptor (FcγR)-dependent mechanisms (Figure 5) [20,25]. In myeloid cells, activating FcγRs and DAP12 represent the principal ITAM-containing signaling adapters [12,14]. These adaptors provide docking sites for kinases, thereby amplifying intracellular signaling cascades [14,29]. The cytoplasmic DAP12 ITAM unit is crucial for functional integrity of cell surface receptors such as TREMs who lack signaling motifs [17]. Like DAP12, TREMs are expressed on diverse myeloid subsets, including granulocytes, monocytes, and tissue macrophages, and have been implicated in both inflammation and autoimmunity [17]. We therefore examined the expression of two key members of this family, TREM1 and TREM2, both of which are present in humans and mice. Upon TREM crosslinking, the tyrosine residues within the ITAM motif of DAP12 become phosphorylated, leading to the recruitment of SYK tyrosine kinases and subsequent activation of downstream signaling cascades, including PI3K [17,34]. Notably, TREM1 and TREM2 appear to exert largely opposing functions with TREM1 acting as an amplifier of inflammation and TREM2 eliciting anti-inflammatory responses [16,35]; however, both receptors can also exert similar inflammatory responses, such as in the gut environment, and their functions may vary depending on the cell type involved [17]. In our study, TREM1 expression was strongly reduced in DAP12^−/−^ skin, but increased in wildtype BP skin, yet this did not translate into functional differences in disease outcome. This finding is consistent with prior work demonstrating that altered TREM1 expression in inflamed skin has no significant impact on the pathogenesis of cutaneous disorders, such as psoriasiform dermatitis, contact dermatitis, and epidermolysis bullosa acquisita (EBA) [36]. It further supports the view that TREM1 function is primarily driven by microbial rather than sterile inflammatory stimuli [17,35,37]. Similarly, TREM2 expression was markedly reduced in anti-Col17 IgG-treated wildtype mice as well as in DAP12^−/−^ mice compared with untreated control skin, but this reduction also appeared irrelevant to BP pathogenesis. These data align with the notion that TREM2 functions mainly in lipid homeostasis and hair growth rather than in autoantibody-driven inflammation [17,38,39].

In addition to myeloid cells, DAP12 is expressed on various lymphocyte subsets, including natural killer cells and some T cell populations [10,11,12]. We therefore assessed the immune cells composition in BP and control skin. Antibody-transfer models of BP, including the anti-Col17 IgG-mediated BP, are typically characterized by predominant neutrophil infiltration [20,25,40,41]. While neutrophil frequencies were comparable across lesional skin biopsies, interestingly eosinophil infiltration was significantly reduced in anti-Col17 IgG-treated DAP12^−/−^ mice compared to wildtype BP controls, suggesting that DAP12 may influence eosinophil biology. However, eosinophils are not the major effector cells in murine BP models, in contrast to human disease where they may play a more prominent pathogenic role [7,20,25,40,41]. Other immune cells subsets, including T cells, dendritic cells, and natural killer cells, were altered only by antibody treatment itself and were unaffected by DAP12 deficiency.

Lastly, we investigated the pharmacological modulation of PI3K, focusing on the δ isoform, using parsaclisib. PI3Kδ signaling has been linked to both the FcγR and C5a/C5aR1 pathways (Figure 5). Specifically, PI3Kδ is essential for FcγR-mediated in immune complex-driven inflammation and contributes to C5a-induced impairment of neutrophil phagocytosis [42,43,44]. In addition, PI3Kδ has been shown to promote TREM1-dependent neutrophil activation [45]. Other studies further demonstrated that TREM1 and TREM2 signaling via DAP12 leads to recruitment of SYK and subsequent activation of different PI3K isoforms [17]. Consistent with these observations, pharmacological inhibition of PI3Kδ with LAS191954 and/or parsaclisib improved outcomes in several pemphigoid models, including antibody transfer- and immunization-induced EBA as well as antibody transfer-induced mucous membrane pemphigoid [18,46]. However, in anti-Col17 IgG-mediated BP, parsaclisib treatment did not ameliorate disease activity, aligning with prior findings in antibody transfer-induced EBA [18]. These contrasting results across different pemphigoid mouse models were partly attributed to distinct kinome signatures in lesional skin and oral mucosa. Attenuated disease progression was observed only in models where PI3Kδ was integrated into the kinome activation network, particularly those dominated by cGMP-/cAMP-dependent protein kinases (immunization-induced EBA) as well as the SYK/Src kinases (mucous membrane pemphigoid) [18]. Furthermore, differences in therapeutic efficacy between different PI3Kδ inhibitors may reflect distinct in vivo pharmacokinetics and drug-specific effects [18]. For example, LAS191954 was shown to reduce IL-8-induced polymorphonuclear leukocyte migration, whereas such an effect was not observed with parsaclisib [18,46].

## 5. Conclusions

Taken together, our in vivo findings indicate that although DAP12 signaling influences local immune cell composition, the DAP12/TREM axis is dispensable for BP development and does not appear to be a promising therapeutic target for this disease. Nonetheless, it is important to note that the BP mouse model, while recapitulating key clinical and immunopathological features of human BP, is limited to the effector phase of the disease and may not capture the full spectrum of immune responses underlying human disease. Moreover, the DAP12/TREM pathway might play a more prominent role in other pemphigoid diseases, such as mucous membrane pemphigoid, which partially involve distinct autoantigens and kinome activation patterns more amenable to DAP12-dependent signaling.

## Figures and Tables

**Figure 1 biomolecules-15-01549-f001:**
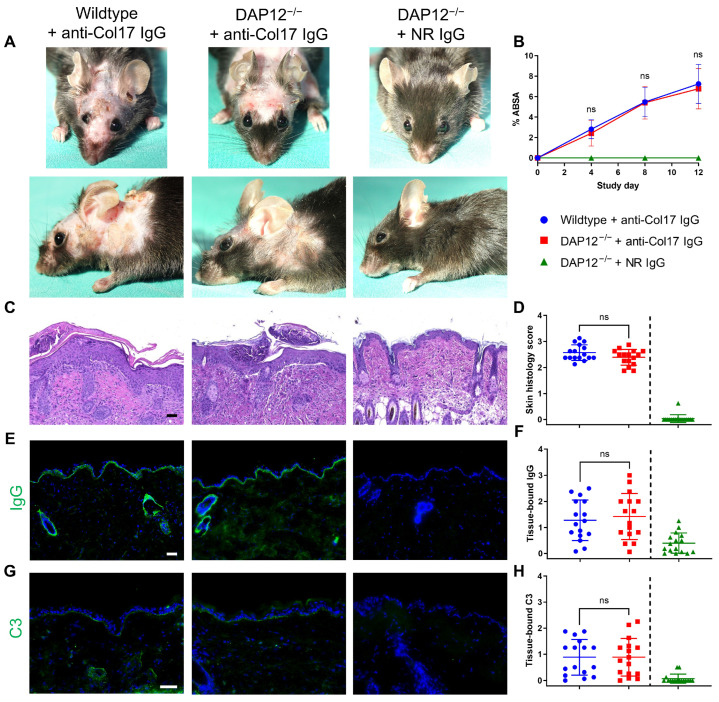
DAP12 deficiency does not alter disease activity in anti-type XVII collagen (Col17) IgG-induced bullous pemphigoid. (**A**,**B**) Clinical manifestations and affected body surface area (ABSA) in DAP12^−/−^ and C57BL/6J wildtype mice (two-way ANOVA with Sidak’s multiple comparison test; *n* = 17/group). (**C**,**D**) Hematoxylin & eosin staining and histopathological assessment of lesional and site-matched control skin (Mann–Whitney test; *n* = 16/group). (**E**–**H**) Deposition of IgG and C3 along the dermal-epidermal junction and corresponding semiquantitative scores (Mann–Whitney test; *n* = 16/group). Scale bars = 50 μm. DAP12^−/−^ mice receiving normal rabbit (NR) IgG were excluded from all analyses. ns, not significant. Data are presented as mean ± standard deviation.

**Figure 2 biomolecules-15-01549-f002:**
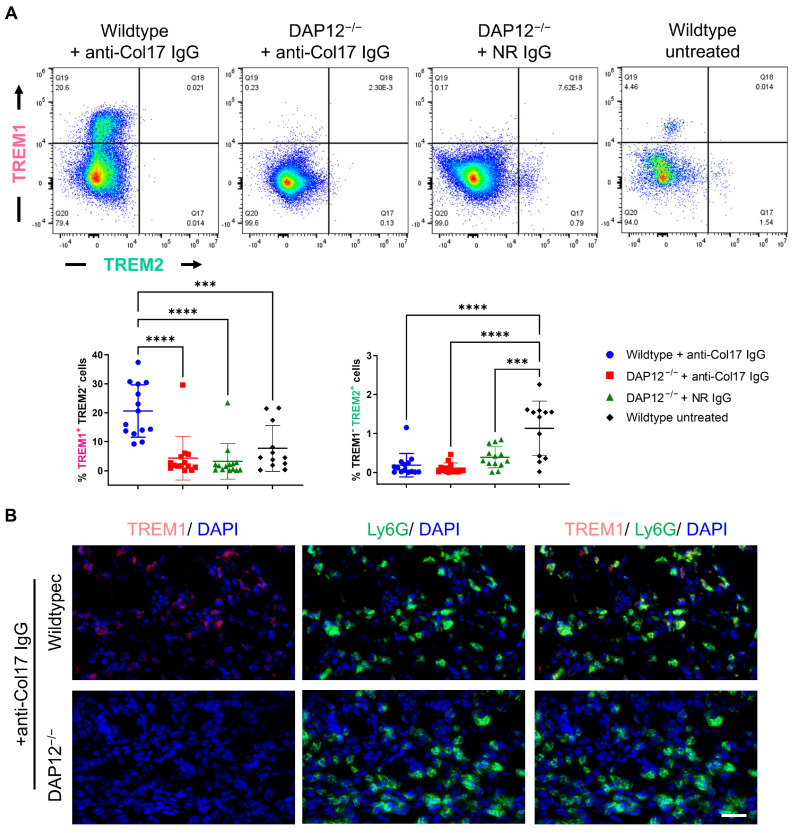
Dysregulation of DAP12-associating receptors TREM1 and TREM2 does not influence bullous pemphigoid pathogenesis. (**A**) Flow cytometry analysis of TREM1 and TREM2 expression in lesional skin biopsies and site-matched control skin (one-way ANOVA with Tukey’s multiple comparison test; ****, *p* < 0.0001; ***, *p* < 0.001; *n* ≤ 14/group). (**B**) Indirect immunofluorescence staining of TREM1 and Ly6G in lesional skin biopsies. Scale bar = 40 μm. Data are presented as mean ± standard deviation.

**Figure 3 biomolecules-15-01549-f003:**
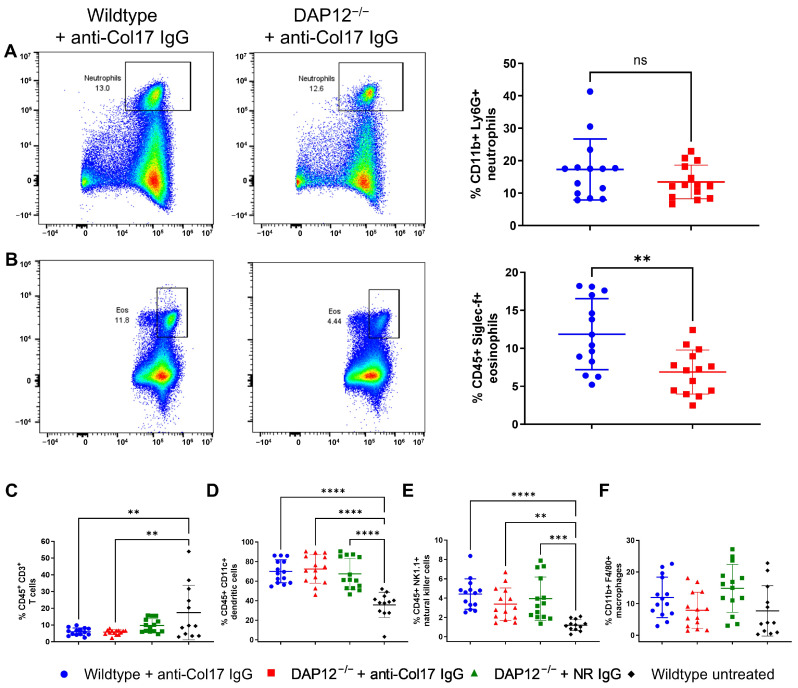
Changes in the composition of inflammatory infiltrates in DAP12-deficient skin do not affect bullous pemphigoid development. (**A**,**B**) Flow cytometric analysis of CD11b^+^Ly6G^+^ neutrophils and Siglec-f^+^ eosinophils in lesional skin from anti-type XVII collagen (Col17) IgG-treated wildtype and DAP12^−/−^ mice (Mann–Whitney test; *n* = 14/group). (**C**–**F**) Flow cytometric analysis of skin resident CD3^+^ T cells, CD11c^+^ dendritic cells, NK1.1^+^ natural killer cells, and CD11b^+^F4/80^+^ macrophages in lesional skin from anti-Col17 IgG-treated mice and site-matched control skin from normal rabbit (NR) IgG-treated and untreated mice (one-way ANOVA with Tukey’s multiple comparison test; *n* ≤ 14/group; only significant changes are indicated). ns, not significant. **** *p* < 0.0001; *** *p* < 0.001; ** *p* < 0.01. Data are presented as mean ± standard deviation.

**Figure 4 biomolecules-15-01549-f004:**
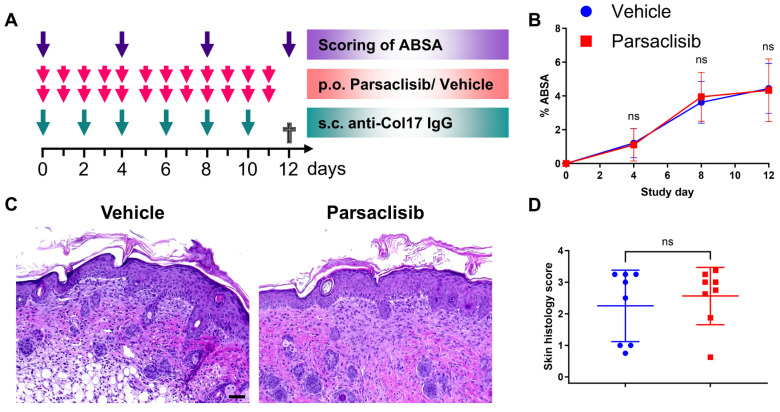
Inhibition of the PI3Kδ isoform does not alter disease activity in anti-Col17 IgG-induced bullous pemphigoid. (**A**) Study design. (**B**) Affected body surface area (ABSA) in parsaclisib vs. vehicle-treated mice (two-way ANOVA with Sidak’s multiple comparison test; *n* = 16/group). (**C**,**D**) Hematoxylin & eosin staining and histopathological scoring of lesional skin (Mann–Whitney test; *n* = 8/group). Scale bars = 50 μm. ns, not significant; p.o., per os; s.c., subcutaneous. Data are presented as mean ± standard deviation.

**Figure 5 biomolecules-15-01549-f005:**
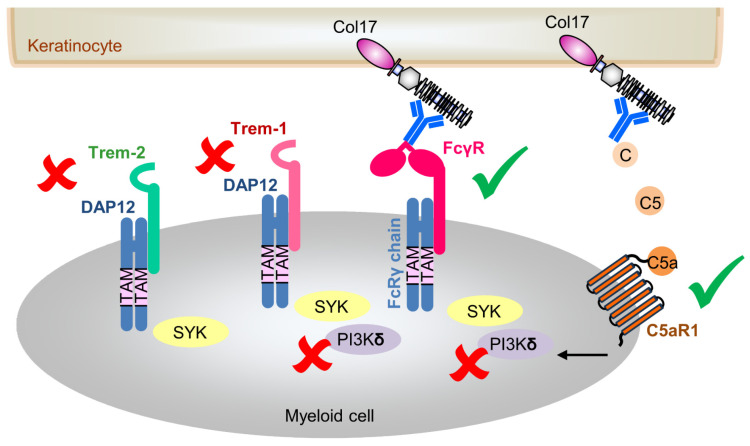
Schematic overview of immunopathogenic mechanisms underlying anti-type XVII collagen (Col17) IgG-induced bullous pemphigoid. The illustration depicts the dispensable DAP12/TREM signaling pathway (marked with red crosses), alongside Fc gamma receptor (FcγR) and complement-mediated effector mechanisms implicated in disease pathogenesis (marked with green check marks).

## Data Availability

The original contributions presented in this study are included in the article. Further inquiries can be directed to the corresponding author.

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
