# Peer review of "Bullous Pemphigoid Develops Independently of DAP12"

_biomolecules, 2025, doi:10.3390/biom15111549_

Round 1
Reviewer 1 Report
Comments and Suggestions for Authors
Review of Manuscript: “Bullous Pemphigoid Develops Independently of DAP12” by Pigors et al.
General Comments
The authors investigate the role of the adaptor molecule DAP12 in the pathogenesis of bullous pemphigoid (BP), the most common autoimmune blistering skin disease. Using an anti-type XVII collagen (Col17) IgG transfer mouse model, they compare disease severity between DAP12-deficient and wild-type mice and explore expression of DAP12-associated receptors TREM1 and TREM2, as well as the impact of PI3Kδ inhibition using parsaclisib.
Minor Issues
1. Abstract
- Line 20, Page 1: “IgG injections over 12 days produced comparable disease activity” >> could specify sample sizes and emphasize that this supports DAP12-independence.
2. Methods
- Section numbering skips from 5 to 2.7; likely a formatting issue (please compare Line 127 and 131, Page 3)
- Specify whether investigators were blinded to genotype during scoring (it is mentioned for disease scoring but should be explicit for histology/IF).
3. Results
- Please add “in text” numerical data for clarity, not only in figure captions.
- In all the “Legends”, add (means ±SD).
Thus, it is stated in “2.8. Statistical Analyses” but it would be optimal also in “Legends” for the readers.
4. References
- Excellent coverage and current citations, including Nat Rev Dis Primers 2025 (Line 366-367).
- Minor formatting inconsistencies (e.g., “immunoreceptor tyrosine-based activation motif (ITAM) domain” could be standardized throughout the manuscript).
My Recommendation: Minor Revision
The manuscript is technically sound, well-structured, and contributes valuable mechanistic insight—even though the results are negative. Minor improvements in data presentation, contextual discussion, and clarity will enhance readability and scientific impact.
Author Response
Minor Issues
- Abstract
- Line 20, Page 1: “IgG injections over 12 days produced comparable disease activity” >> could specify sample sizes and emphasize that this supports DAP12-independence.
Response: The sentence has been revised as follows: “Repeated anti-Col17 IgG injections over 12 days produced comparable disease activity in DAP12-deficient and wildtype mice (n=17/ group), indicating that disease induction occurs independently of DAP12 signaling.”
- Methods
- Section numbering skips from 5 to 2.7; likely a formatting issue (please compare Line 127 and 131, Page 3)
Response: Thank you for pointing out the error in the section numbering. The numbering was revised accordingly.
- Specify whether investigators were blinded to genotype during scoring (it is mentioned for disease scoring but should be explicit for histology/IF).
Response: Methods sections 2.4 and 2.5 were revised accordingly.
- Results
- Please add “in text” numerical data for clarity, not only in figure captions.
Response: We have added sample sizes and p values to the respective Results sections.
- In all the “Legends”, add (means ±SD). Thus, it is stated in “2.8. Statistical Analyses” but it would be optimal also in “Legends” for the readers.
Response: We have included the following statement in the legends of Figures 1–4: “Data are presented as mean ± standard deviation.”
- References
- Excellent coverage and current citations, including Nat Rev Dis Primers 2025 (Line 366-367).
Response: Thank you for your positive feedback. We are pleased that you found the choice of citations appropriate.
- Minor formatting inconsistencies (e.g., “immunoreceptor tyrosine-based activation motif (ITAM) domain” could be standardized throughout the manuscript).
Response: Formatting inconsistencies have been revised accordingly.
Reviewer 2 Report
Comments and Suggestions for Authors
The role of DNAX-activating protein of 12 kDa (DAP12) in autoimmune diseases are currently controversial. To clarify this issue, in this study, the authors examined the pathogenic role of DAP12 in the mouse model for bullous pemphigoid (BP), in which the disease is induced by the injection of antibodies to BP180 NC14-1 domain. The DAP12-knockout mouse and DAP12 inhibitor were used, and the expression levels of Triggering receptor expressed on myeloid cells (TREM)-1/2 and phosphatidylinositol 3-kinase (PI3K) were examined. The results indicated that, although DAP12 signaling modulates local immune cell composition, DAP12/ TREM1/2-axis is dispensable in the development of skin disease in this mouse model. This result may also indicate the more important pathogenic role for FcgR and complement cascade in the pathogenesis in BP.
This is a unique and important study, although this study showed rather negative data. Therefore, more studies using different systems may be needed in the future to address this question.
This is well designed study, and the experiments were properly performed. The manuscript is well written, and figures and English are good. However, I have a few comments, which are described below.
(1) The authors used the new mouse model for BP, which the authors recently reported on 2024 (reference 20). Because the previous studies indicated that BP180 NC16 (NC15a in mouse) domain is pathogenic and the sequences of this region are quite different between mouse and human, leading to the difficulty to develop mouse model for BP. In the mouse model used in this study, the authors used antibodies to NC14-1 domain (C-terminal domain) of BP180, which may not be pathogenic. Therefore, to better understanding of the readers, the authors should briefly mention why this mouse model can develop the skin lesions, and how this model is different from the previous mouse models in the earlier part of the manuscript.
(2) The full spell for the abbreviation “TREM” may not be correctly shown at line 22 in the abstract section and at line 66 in the introduction section.
(3) The authors should exchange “two groups” at line 154 and “DAP12 -/- and wild type mice” at line 156 for the better understanding of the readers
(4) The figure legend for the figure 3 is shown in both pages 7 and 8. This structure for the figure 3 and its figure legend should be corrected.
Author Response
(1) The authors used the new mouse model for BP, which the authors recently reported on 2024 (reference 20). Because the previous studies indicated that BP180 NC16 (NC15a in mouse) domain is pathogenic and the sequences of this region are quite different between mouse and human, leading to the difficulty to develop mouse model for BP. In the mouse model used in this study, the authors used antibodies to NC14-1 domain (C-terminal domain) of BP180, which may not be pathogenic. Therefore, to better understanding of the readers, the authors should briefly mention why this mouse model can develop the skin lesions, and how this model is different from the previous mouse models in the earlier part of the manuscript.
Response: We thank the reviewer for this valuable comment. We have added the following text to the Introduction: “In this study, we investigated the role of the DAP12 axis in a murine model of BP induced by transfer of anti-Col17 IgG targeting the murine extracellular non-collagenous domains 14–1 (NC14–1), which, in addition to IgG targeting the NC15A domain, the murine homologue of the human immunodominant NC16A domain, are capable of triggering an inflammatory response and tissue injury in the skin.”
(2) The full spell for the abbreviation “TREM” may not be correctly shown at line 22 in the abstract section and at line 66 in the introduction section.
Response: Thank you for pointing the inconsistencies in using the abbreviation for “TREM”. The text in the Abstract and Introduction has been changed accordingly.
(3) The authors should exchange “two groups” at line 154 and “DAP12 -/- and wild type mice” at line 156 for the better understanding of the readers
Response: We agree that an exchange of the phrases will improve readability and clarity and have revised the text accordingly. “Throughout the study period, disease activity, as assessed by body surface area involvement, was comparable between the DAP12-/- and wildtype mice (n=17/ group) (Figure 1A,B). Similarly, histopathological features of lesional skin biopsies, including subepidermal blister formation and inflammatory cell infiltration, were comparable between the two groups (n=16/ group)”
(4) The figure legend for the figure 3 is shown in both pages 7 and 8. This structure for the figure 3 and its figure legend should be corrected.
Response: The formatting has been corrected and the legend for Figure 3 is now shown on page 8 only.
Reviewer 3 Report
Comments and Suggestions for Authors
The manuscript “Bullous Pemphigoid Develops Independently of DAP12” is, in my opinion, a well-written and methodologically solid experimental paper. It presents a clearly defined research question—evaluating the role of DAP12 signaling in autoimmune bullous pemphigoid—and provides valid and well-supported conclusions. The topic is relevant and timely, addressing an interesting gap concerning innate immune signaling pathways in autoimmune skin disease.
The study design is rigorous, with appropriate use of DAP12-deficient mice, clearly described controls, and validated outcome measures such as histology, flow cytometry, and immunofluorescence. The conclusions are consistent with the presented data and align with previous work in related autoimmune models. The figures and tables are clear and adequately support the narrative. The references are current and relevant, including key mechanistic and methodological citations.
Minor editorial or stylistic suggestions can be addressed during copyediting. In summary, this is a scientifically sound and well-executed study suitable for publication after standard editorial review.
Major Comment
- Introduction:
I recommend adding a schematic figure illustrating the DAP12/TREM1/2, FcγR, and PI3Kδ signaling cascades. This would greatly improve clarity for readers unfamiliar with immunoreceptor signaling and facilitate understanding of the mechanistic interactions described later in the Results and Discussion sections.
Author Response
Major Comment
- Introduction:
I recommend adding a schematic figure illustrating the DAP12/TREM1/2, FcγR, and PI3Kδ signaling cascades. This would greatly improve clarity for readers unfamiliar with immunoreceptor signaling and facilitate understanding of the mechanistic interactions described later in the Results and Discussion sections.
Response: We thank the reviewer for this valuable comment. We would like to note that a schematic illustration of the DAP12/TREM1/2, FcγR, and PI3Kδ signaling cascades is provided in Figure 5 of the manuscript. We believe this Figure is most effective as a summary at the end of the manuscript; however, if the reviewer and editorial team prefer to present it at the beginning, as part of the Introduction, we would be happy to accommodate this change.
Round 2
Reviewer 2 Report
Comments and Suggestions for Authors
None.